# Combining N-mixture and occupancy analysis offers a more complete picture of carnivore habitat use in Northeastern Türkiye

**J. David Blount**[1]*, **Austin M. Green**[2], **Mark Chynoweth**[3], **Emrah Çoban**[4], **Josip Kusak**[5], **Çağan H. Şekercioğlu**[1,4,6]

**1** School of Biological Sciences, University of Utah, Salt Lake City, Utah, United States of America,, **2** Science Research Initiative, University of Utah, Salt Lake City, Utah, United States of America, **3** Department of Wildland Resources, Utah State University, Vernal, Utah, United States of America, **4** KuzeyDoğa Society, Ortakapı Mah. Şehit Yusuf Cad. No: 93 Kat: 1, Merkez, Kars, Türkiye, **5** Faculty of Veterinary Medicine, University of Zagreb, Zagreb, Croatia, **6** Department of Molecular Biology and Genetics, Koç University, Istanbul, Türkiye

* David.Blount@utah.edu.edu

## Abstract

Occupancy and N-mixture analyses have been successfully used to understand habitat use in various species. However, since these methods fundamentally answer different questions about wildlife distribution, the results from each modelling approach may provide different insights into species' habitat use. In this study, we leveraged data from a long-term camera trapping study in northeastern Türkiye to compare the results from occupancy and N-mixture analyses, with the main objective of understanding how the modelling approach used can influence our knowledge of species' habitat use. Specifically, we compared the habitat use preferences from N-mixture and occupancy analyses for three carnivore species with varying baseline abundances. Our results provide evidence that occupancy and N-mixture analyses provide different insights into species' sensitivity to environmental and anthropogenic factors. Whereas occupancy analysis provides a relatively broad summary of the factors that affect where a species may or may not be located on a landscape or which areas they may be more likely to use over a certain time period, N-mixture modelling may provide insights into the factors that affect the degree of use at individual sites, with particular emphasis on being able to deduce small-scale changes in habitat use across a landscape. Furthermore, while the detection probability of an occupancy analysis has been formally used as a measure of site use intensity, N-Mixture models may offer higher resolution of the quantity of use. Therefore, as these two methods tend to investigate habitat use at different spatial scales, when used in conjunction they can provide a more refined understanding of species' habitat use through repeat-survey sampling methods like camera trapping.

**Data availability statement:** All data are available from Zenodo at DOI 10.5281/zenodo.14618278.

**Funding:** We are grateful to Fondation Segré and the Sigrid Rausing Trust for providing the majority of the funding for this project. This research was also supported by other generous donors, including Arkadaşlar, Bilge Bahar, Seha İşmen, Ömer Külahçıoğlu, Burak Över, Batubay Özkan, Emin Özgür, Suna Reyent, Ceren Sağlamer, Faruk Yalçın Zoo, National Geographic Society, STGM, TANAP, TÜBİTAK, Barbara Watkins, and the Whitley Fund. The funders had no role in study design, data collection and analysis, decision to publish, or preparation of the manuscript.

**Competing interests:** No authors have competing interests.

## Introduction

Species' habitat use can be influenced by many factors. Understanding the variables that affect species' habitat use can provide insights into the effects of competition, fitness, diet, and anthropogenic disturbance [1–6]. At large scales, habitat use models can help predict species' distributions [4,7], while at finer scales, habitat use models can identify areas of significant conservation concern or heightened human-wildlife conflict [8,9]. To understand habitat use, researchers have developed multiple modelling methodologies. For example, occupancy [10] and N-mixture models [11–13] are two commonly used statistical frameworks that can quantify habitat use while also accounting for the vagaries of sampling (e.g., imperfect detection).

Occupancy and N-mixture models answer two distinct questions about species' habitat use, but each has been used to assess species' habitat use broadly [10–12]. Occupancy modeling predicts the presence/absence of species while accounting for imperfect detection across sites using detection/non-detection data [13–15]. Occupancy models are widely used to assess habitat use across multiple taxa [11,14,16–18]. While occupancy modeling predicts species' habitat use through detection/non-detection data, N-mixture modeling can provide insights into detection-corrected site use *intensity,* or the relative amount of activity at a site, by utilizing repeated counts instead of detection/non-detection data [10,19–22].

While both occupancy and N-mixture models have been successfully adapted to predict habitat use, both models come with drawbacks. Occupancy models can be less sensitive when species are rare and wide-ranging or common and ubiquitous across study sites, as using detection/non-detection data can make it difficult to comparatively assess the relative intensity at which individual sites are used (i.e., boundary estimation issues) [14,23–25]. On the other hand, N-mixture models utilize count data, which may better assess relative activity across sites, but come with issues such as estimation instability and high computational demand [26]. Currently, there are only a few studies successfully comparing the results of these modelling frameworks on the same dataset, and none focusing on mammal systems [27,28].

In this study, we used a 13-year camera trapping dataset from northeastern Türkiye to compare the differences in modeled habitat use of three carnivore species through N-mixture and occupancy analyses. We aimed to answer two major questions: 1) what is the difference in habitat use across carnivore species, and 2) how do the results from N-mixture and occupancy modelling compare across species? By answering these two questions, we hope to shed light on the habitat use of three carnivores in a region with very little published data and provide guidance on the use of these two methods.

## Methods and materials

### Study site

This study took place on the Kars-Ardahan high plateau in northeastern Türkiye (Fig 1). The core study area (~550 km$^2$; 40°20'N 42°35'E) includes the Sarıkamış-Allahuekber Mountains National Park (hereafter SAMNP) and the surrounding forests. Central to

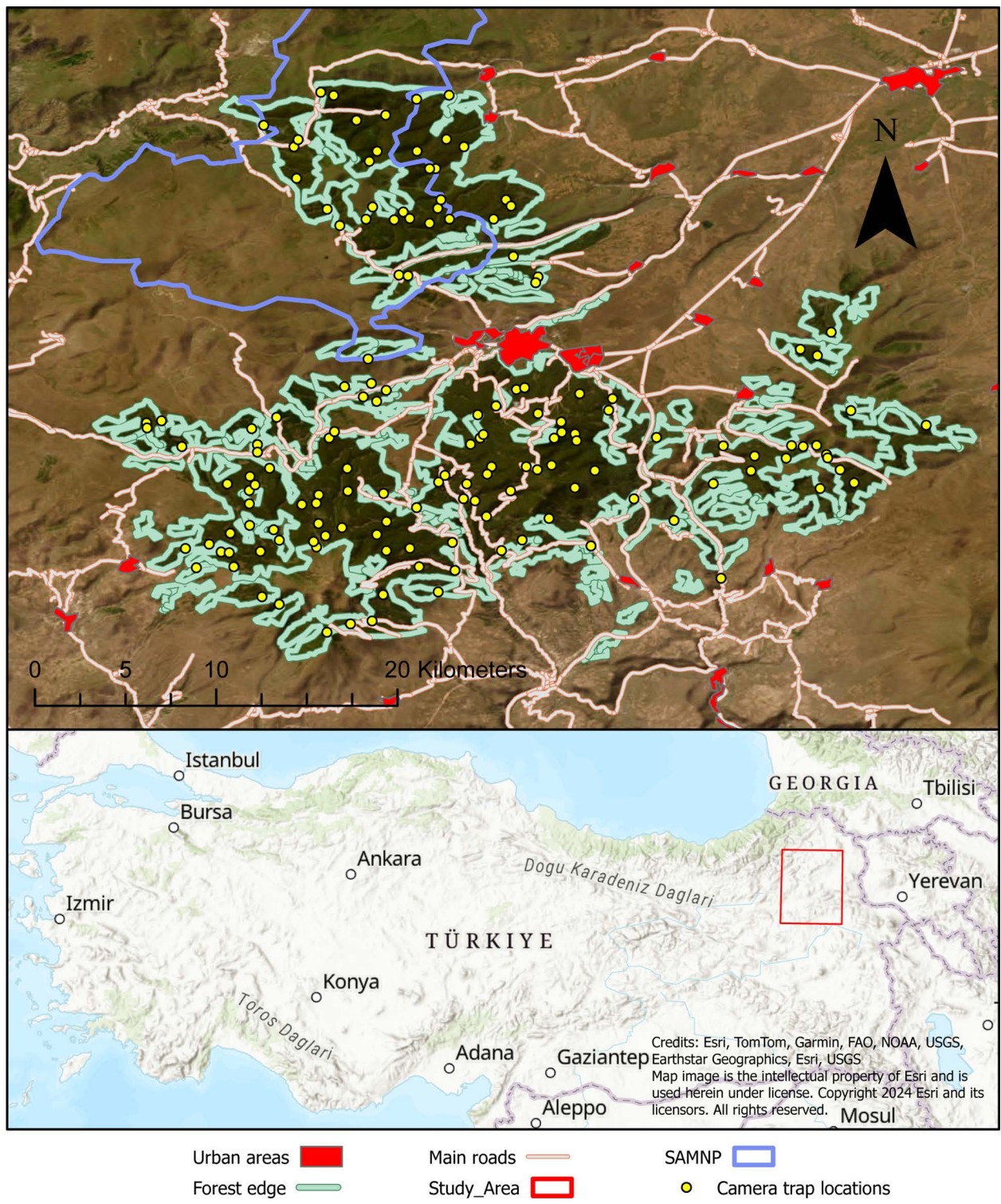

**Fig 1. Study area location, camera trap locations, national park boundary, urban areas, and major roads.** In the map above, the blue outline represents the boundaries of the SAMNP, with urban areas in red. Yellow circles represent camera trap locations, and major roads are shown in light red. The light green line represents the boundaries of the forested region where all cameras were located. In the lower map, the red outline represents the outline of the study area.

the study area is the city of Sarıkamış (population: c.15,600 in 2023; [29]). The SAMNP is located north of the city, while the remainder of the forest is located to the south, divided from the SAMNP by a major road (Fig 1). The SAMNP covers an area of 225.2 km$^2$, but only 22.07% or 49.69 km$^2$ is forested [30]. The total study area includes 328.38 km$^2$ of forested area [31]. Most of the forest cover is Scots pine (*Pinus sylvestris),* accompanied by small patches of aspen (*Populus tremula*). Under-story vegetation is scarce due to extensive grazing, firewood collection, and intensive human intervention. The study area ranges in elevation from 1,900–3,120 m asl and is composed of fragmented forest in a matrix of agriculture and rangelands [31]. The climate is continental, with temperate summer months from June through September (monthly average 13° to 18° C) and cold winter months with snowfall from November through March (monthly average -10° to 0° C;[32]). Outside the harsh winter months, human activity is a dominant presence in both time and space. Anthropogenic activity in the forest con-sists of livestock grazing, legal and illegal timber harvest, the collecting of non-timber forest products (NTFP) and recreation that includes a ski resort. Livestock in the area is composed of cattle, sheep, and goats, freely roaming in rangelands and pastures from April to November [30]. An unfenced town dump is located 6 km west of the city of Sarıkamış and represents an important food source for bears [33] and a minor food source for wolves [30]. Specifically, some bears in the area exhibit altered life histories to regularly use the dump, while others never visit it [33].

Three carnivore species occur within the study site, each at a different abundance [31]. Winter tracking surveys have shown that the population of gray wolves (*Canis lupus;* hereafter referred to as wolves) exists at a density of 3.0 wolves per 100 km$^2$ (Range:1.2 to 6.4; [34]), with the most recent snowtracking efforts estimating a population size of 28 individ-uals, while that varies through the years (Kusak and Şekercioğlu, unpublished data). While Eurasian brown bear (*Ursus arctos arctos;* hereafter referred to as bear) density has never been estimated within the study area, they are considered far more abundant than the other two carnivores. Specifically, 59 individual bears have been seen at the same time at the town dump (Blount pers. obs.), with roughly 40% of the population never visiting the dump [33]. Genetic data indicates the presence of at least 91 brown bears in the region [35]. Finally, Caucasian lynx (*Lynx lynx dinniki;* hereafter referred to as lynx) is thought to exist at the lowest density of 2.1 lynx per 100 km$^2$ based on home range size [36].

## Field work

**Field work was conducted with research permits from Türkiye's Department of Nature.** Conservation and National Parks and the Ministry of Agriculture and Forestry, permit numbers E-21264211-288.04-1602322 and 72784983-488.04-114100. No animals were handled or manipulated over the course of this study.

**Camera trap assays.** Camera traps were deployed between 2010 and 2023. The study site was segmented into 2 km$^2$ grid cells. A combination of Keepguard and Reconyx cameras (Keepguard KG891, Reconyx PC900, Reconyx HPX2) were used. For each sampling session, grid cells were randomly chosen, and cameras were deployed within these selected grid cells along landscape features where animals were most likely to be seen (e.g., dirt roads) [37]. This ensured that cameras would be placed throughout the entire study system and minimized human bias. Cameras were placed on trees at a height of approximately 3 m, 1.5 m or 0.3 m (canopy, chest, or knee height) and pointed towards the road. Cameras were left in the field for an average of 111.2 days (range: 1–728 days) before they were moved to another randomly selected grid cell. Cameras were occasionally placed in the same locations and no more than one camera was active in each location at any given time. Locations within 15 m of each other were considered as the same location. Since human and vehicle traffic may change over time, each deployment was counted independently. To reduce the effects of spatial autocorrelation while maximizing the amount of data we retained, we chose to cap the number of deployments at a single site to four deployments. If there were more than four deployments at a given site, we chose the four deployments with the longest duration. Furthermore, only one deployment was recorded per site per year. Camera sites were located at an average of 886 m (range 183–3745 m) from each other. Cameras were programmed to take photos with no reset interval between triggers, and the sensitivity was set to "high." Each camera was set to operate 24 hours a day. No bait was used at any sites. GPS locations of each camera trap were collected during the time of placement, and environmental

covariates were then extracted for each of the GPS locations using ArcGIS Pro [38]. Height and camera type were recorded at the time of installation. After recovery, the field of view was assessed for each camera and given a score from one to five. A score of three would represent the camera's view being able to capture a full vehicle, a score of five representing the camera being able to see multiple car lengths, and a score of one representing a field of view less than one car length. To understand the optimal time interval between photos to account for independent events, we identified the time between events for the same species on the at the same site. We then looked through each paired event (the first animal and the subsequent animal) to identify whether it was the same animal in both events or if they were different. We used body size, color, shape, direction of travel, and personal judgment to assess if both events were recording the same individual. We then chose a time point for each species that would avoid double counting while limiting undercounting new individuals. This time period was decided to be 10 minutes for bears, one minute for lynx, and 30 minutes for wolves. We expanded the time period for wolves so each event would include the entire pack. Failure to include the entire pack could bias the N-mixture models to report higher habitat use selected for by larger packs rather than sites that are used more frequently across packs. The sampling occasion was set at seven days, which was the minimum time between the same collared individual appearing twice within the camera array.

## Statistical analyses

**Covariate selection.** To assess habitat use across a range of carnivore species, we chose topographic and landscape covariates that have been shown to effect the habitat use of our species of interest (Table 1). Topographic features like elevation, ruggedness, and slope have been shown to affect habitat use in wolves [39–41], lynx [42,43], and bears [40,44,45]. All features were calculated from the ESRI Terrain layer in ArcGIS. Landscape features like forest cover, distance to main roads, and distance to urban areas have also been shown to have significant effects on the habitat use of these large carnivores [39,40,42,44]. Forest cover was estimated using a radius of 500 m, 1 km, and 10 km scales and is defined as the percent forest cover in each of these three buffers surrounding each camera site [46–48]. There are several main roads that bisect the study area (Fig 1). Euclidian distance was calculated from each of these roads to each of the camera sites in meters. This same approach was done to calculate distance from each camera site to urban areas (Fig 1) and distance from the dump. The town dump, that is six km outside of Sarıkamış, is a major food source for bears and many bears access this food source daily [30,31]. We used distance to dump as a covariate to understand how this anthropogenic food resource may affect inter and intra-specific interactions between bears and the other carnivores. Finally, people use the forests for natural resources like timber, NTFPs and for grazing livestock. Humans and vehicles may be perceived by carnivores as different in their risk as vehicles pass through areas quickly, but humans are typically accompanied by both dogs and livestock. Therefore, we calculated the average daily detection rate of vehicles and humans separately for each camera.

A covariate describing year was intentionally left out of the model to avoid overparameterizing our model, and it was not a covariate of interest for this study. We did control for sites with more than one deployment with the addition of replicate level intercept in the detection of both models. All covariates were assessed for intercorrelation using a Person's correlation test. Only four covariates, ruggedness, slope, 1 km forest cover, and 10 km forest cover were highly intercorrelated (Person's $|r| > .7$). A univariate analysis was run to compare slope and ruggedness, and the effect of slope was greater than that of ruggedness for all species. Therefore, slope was included and ruggedness was excluded from the final analysis. Since 500 m forest cover was not highly correlated with any other covariate, we excluded 1 km and 10 km forest cover covariates. Our model consisted of additive effects from the remainder of the covariates listed in Table 1. All covariates were standardized with a mean of 0 and a standard deviation of 1, which allowed for a more direct comparison of effect size across predictors and aided in the numerical optimization of modeled parameters. Field of view, camera type, and height of camera were included as observational level covariates within both models but returned non-sensible results (all factors having a negative effect) and were excluded from further models.

**Table 1. Description and origin of covariates assessed for use in N-mixture and occupancy analyses. * signifies that that covariate was included in the N-mixture and occupancy models.**

| Covariate | Included in final model | Description | Origin |
|---|---|---|---|
| Distance from Dump | * | Distance of the camera from the town dump. The dump has a significant impact on several species, and wolves and bears have both been noted as feeding here. | Distance from polygon created based on location of dump found in ESRI satellite basemap |
| Forest Cover | * 500m only | Percent of forest within a buffer surrounding each camera trap | Corine land cover 2018 forest areas expanded |
| Distance from Main Roads | * | Distance from the main roads to the camera traps. Main roads were defined as the two large highways that cross the study site. | Roads were mapped across all field seasons with new roads being added as they were found. |
| Distance From Urban Areas | * | Distance to the nearest anthropogenic resource from each camera trap. Anthropogenic resources can be cities, towns, military bases, or restaurants. | Corine land cover 2018 Urban areas expanded to include fine scale urban areas not included in CLC |
| Elevation | * | A digital terrain model of ground heights with 0 = Sea level | ESRI terrain |
| Roughness | | The amount of elevation difference between adjacent cells | ESRI terrain with TRI toolbox |
| Slope | * | Derived from ESRI terrain model, measured in degrees of slope | ESRI terrain |
| Human Activity | * | Trap rate of humans within each camera trap deployment | Calculated at each site from camera trap data |
| Vehicle activity | * | Trap rate of vehicles within each camera trap deployment | Calculated at each site from camera trap data |

**N-mixture Models.** For N-mixture models, we modified a Bayesian co-abundance N-mixture model [21] into a multi-species, single-year model. This model was chosen over other N-mixture approaches as it allowed us to control for overdispersion in estimated counts per site [10]. Specifically, we modeled local abundance (expected count) $n_{i,k}$, at site $i$ in 1, …,I sites and for $k$ in 1, …, K species as

$$n_{i,k} \sim Poisson(\lambda_{i.k})$$

where $\lambda_{i.k}$ denotes the expected number of individuals of species $k$ detected per sampling occasion at site $i$. We then modeled the effects of anthropogenic and environmental factors on each species $k$ as

$$\log(\lambda_{i.k}) = \alpha_k x_i$$

where $\alpha_k$ is a vector of species-specific parameters and $x_i$ is a conformable vector of site-level covariates where the first element is a 1 to account for the model intercept (Table 1).

Imperfect detection in the expected number of individuals present was accounted for by assuming the detection of species $k$ at site $i$ during the sampling occasion $j$ in 1, …, J repeat samples, where $y_{i,j,k}$ follows a binomial distribution

$$y_{i,j,k} \sim Binomial(n_{i,k}, \rho_{i,j,k})$$

and where $\rho_{i,j,k}$ represents the detection probability of species $k$ at site $i$ during the sampling occasion $j$. We then modeled the effects of the covariates on $k$ species' detection probability as:

$$logit(\rho_{i,j,k}) = \beta_{m,k} + \varepsilon_{i,j,k}$$

where $\beta_{m,k}$ represents a replicate and species-specific intercept for m in 1, …, M site replicates and $\varepsilon_{i,j,k}$ represents a species, occasion, and site-specific over dispersion random effect parameter that follows a normal distribution

   

$$\varepsilon_{i,j,k} \sim normal(0, \sigma)$$

where σ is the standard deviation of $\varepsilon_{i,j,k}$.

## Occupancy Model

For our occupancy analysis, we adopted a hierarchical, Bayesian multi-species approach [49, 50] to estimate the probability that each species used the habitat at each site while correcting for imperfect detection [11,51]. Specifically, we modeled occupancy, *z*, at site *i* in 1,…, I sites and for species *k* in 1, …,K species ($z_{i,k}$), as

$$z_{i,k} \sim Bernoulli\ (\psi_{j,k})$$

where $\psi_{j,k}$ denotes the probability species *k* used site *i* (site use probability). We then modeled effects of the anthropogenic and environmental factors on each species *k* as

$$logit\ (\psi_{i,k}) = \phi_k \gamma_i$$

where $\phi_k$ is a vector of species-specific covariate effects included in the model and $\gamma_i$ is a conformable vector of site-level covariates where the first element is a 1 to account for the model intercept (Table 1).

Imperfect detection of species presence was accounted for by assuming the detection of species *k* at site *i* during sampling occasion *j* in 1,…, J occasions, where $y_{i,j,k}$ follows a Bernoulli distribution

$$y_{i,j,k} \sim Bernoulli(\rho_{i,j,k} z_{i,k})$$

and where $\rho_{i,j,k}$ represents the detection probability of species *k* at site *i* during sampling occasion *j*. We then modeled the effects of covariates on *k* species' detection probability as

$$logit\ (\rho_{i,j,k}) = \beta_{m,k}$$

where $\beta_{m,k}$ represents a replicate and species-specific intercept.

Species-specific parameters were all linked to the community hyperparameters for both N-mixture and occupancy models using the hierarchical approach of Rich et al. [50]. These hyperparameters measure the mean responses of all three species to each of the modeled covariates. Species-specific responses were modeled as normal random effects derived from the hyperparameters [50,52,53].

## Model fit and assessment

All model parameters for both N-mixture and occupancy were estimated within a Bayesian framework using Markov Chain Monte Carlo (MCMC) methods in the program R using the package "jagsUI' [54]. For the occupancy analysis, we ran three chains in parallel, each consisting of 400,000 iterations. For each chain, 200,000 iterations were discarded as burn-in, following an adaptation phase of 200,000, iterations for a total of 600,000 total samples from the joint posterior. We needed fewer iterations for the N-mixture model to converge and used 3 chains of 200,000 iterations following a burn in of 100,000 iterations and an adaptation period of 100,000 iterations. This resulted in 300,000 total samples from the joint posterior. We used diffuse uniform priors for effort and hyper priors for all species-specific parameters. Convergence was calculated using the Gelman-Rubin statistic, where a value less than 1.1 was indicative of successful convergence. This was combined with a visual inspection of parameter trace plots to ensure successful convergence. Finally, model fit was assessed using a Bayesian p-value and Ĉ statistic for the N-mixture model and a Bayesian p-value for the occupancy

model. Bayesian p-values and Ĉ statistics were calculated from simulated and observed data [10,55], where Bayesian p-values between 0.25 and 0.75 indicate a good fit and a Ĉ less than 1.1 suggests minimal unaccounted overdispersion [56,57]. Both values were calculated using $\chi^2$ discrepancy statistics [10,55,58].

We assessed "significant" habitat associations at two levels using the posterior mean and whether the one-tailed 95% or 85% Bayesian Credible Interval (CI) overlapped zero. If zero was not included in the 95% CI, we interpreted this as a "strongly significant" association, and if the 85% CI did not overlap zero, we interpreted this as a "moderately significant" association. The 85% confidence interval is included to capture species and community effects that may be limited by the use of hyperparameters, which can 'shrink' species-specific effects toward the community mean [52,55].

## Results

### Survey efforts

For this study, we used data from 151 distinct sites with 229 deployments, representing 25,471 survey days. In total, we recorded 1,530 detections of the three carnivore species of interest. Bears were observed most frequently, with 688 detections, followed by wolves with 652 detections and lynx with 190 detections. Of these sites, six were sampled four times, 10 sites sampled three times, 28 sites sampled twice, and 107 sites sampled only once.

### N-mixture Model

**Fit.** The N-mixture model had acceptable fit to the data, with a Bayesian p-value of 0.65 and a minimal amount of overdispersion left unaccounted for with a Ĉ value of 0.97. Furthermore, model convergence was acceptable for all monitored parameters for all N-mixture models, with Gelman-Rubin statistics having a mean of 1 and a maximum of 1.07.

**Abundance and detection probability.** The expected count ($n_{i,k}$) for bears was greatest (2.94 bears/site) followed by wolves (2.20 wolves/site) and lynx (1.06 lynx/site; Table 2, Fig 2). For brown bears, detection probability varied between 0.01 and 0.04 (Table 2, Fig 3) between site replicates. This pattern was also found in wolves and lynx whose detection probabilities varied between 0.01 and 0.03. The expected counts and detection probabilities presented here were back-transformed from the reported values produced in our models.

### Occupancy model

**Fit, site use probability, and detection probability.** The occupancy model also had an acceptable fit with a Bayesian p values of 0.49 and model convergence was acceptable for all monitored parameters for the occupancy model. Our Gelman-Rubin statistics had a mean of 1.0 and a maximum of 1.04. (Tables S1-S4). The estimated site use probability was highest for bears (0.96), followed by wolves (0.86), and was the lowest for lynx (0.57; Tables S1-S3). Mean detection probability ranged from 0.09 to 0.18 in bears and was lower on average in wolves (0.08–0.17) and lynx (0.06–0.16).

### Community effects

For both the N-mixture and occupancy analysis, distance from urban area had moderately positive significant effects on site use (mean effect size: 0.47) and expected count (mean effect size: 0.24). For the N-mixture model, the presence of humans had a moderately significant positive effect (mean effect size: 0.34), while forest cover had a moderately significant negative effect (mean effect size: -0.35) on occupancy.

### Habitat Use

Factors that influenced habitat use varied across species and across models. For bears, human presence (mean effect size: 0.42, Fig 4), distance to dump (mean effect size: 0.21), and distance from urban areas (mean effect size: 0.19) were strongly significant positive predictors of habitat use in the N-mixture model. In the occupancy model, distance from urban

**Table 2. The mean expected count and probability with 95% confident intervals in parentheses for each species across each duration window.**

| Species | Model | Parameter | Replicate | Value |
|---|---|---|---|---|
| Bears | N-mixture | Expected Count | N/A | 2.94 (2.46 - 3.56) |
| | | Detection Probability | 1 | 0.02 (0.02 - 0.03) |
| | | | 2 | 0.02 (0.01 - 0.02) |
| | | | 3 | 0.01 (0.01 - 0.02) |
| | | | 4 | 0.04 (0.02 - 0.07) |
| | Occupancy | Site Use Probability | N/A | 0.96 (0.81 - 0.97) |
| | | Detection Probability | 1 | 0.14 (0.13-0.16) |
| | | | 2 | 0.11 (0.09-0.14) |
| | | | 3 | 0.09 (0.06-0.13) |
| | | | 4 | 0.18 (0.12-0.24) |
| Wolves | N-mixture | Expected Count | N/A | 2.20 (1.82 - 2.64) |
| | | Detection Probability | 1 | 0.03 (0.02 - 0.04) |
| | | | 2 | 0.03 (0.02 - 0.05) |
| | | | 3 | 0.01 (0.01 - 0.02) |
| | | | 4 | 0.02 (0.01 - 0.04) |
| | Occupancy | Site Use Probability | N/A | 0.86 (0.76–0.95) |
| | | Detection Probability | 1 | 0.17 (0.15-0.19) |
| | | | 2 | 0.15 (0.12-0.18) |
| | | | 3 | 0.08 (0.05-0.11) |
| | | | 4 | 0.12 (0.08-0.17) |
| Lynx | N-mixture | Expected Count | N/A | 1.06 (0.77 - 1.49) |
| | | Detection Probability | 1 | 0.01 (0.01 - 0.02) |
| | | | 2 | 0.01 (0.00 - 0.02) |
| | | | 3 | 0.02 (0.01 - 0.05) |
| | | | 4 | 0.03 (0.01 - 0.05) |
| | Occupancy | Site Use Probability | N/A | 0.57 (0.42 - 0.75) |
| | | Detection Probability | 1 | 0.06 (0.05–0.08) |
| | | | 2 | 0.07 (0.04–0.10) |
| | | | 3 | 0.12 (0.07–0.19) |
| | | | 4 | 0.16 (0.10–0.22) |

areas (mean effect size: 0.38) was the only shared significant result, which was only moderately significant. The occupancy model also showed that forest cover (mean effect size: -0.43) and slope (mean effect size: -0.36) were moderately significant predictors of habitat use.

Habitat use was most similar between the N-mixture and occupancy models for wolves, with distance to urban areas being significant (N-mixture mean effect size: 0.28, occupancy mean effect size: 0.66, Fig 5). Human activity being strongly significant (N-mixture mean effect size: 0.50, occupancy mean effect size: 1.33) and slope being moderately significant (N-mixture mean effect size: -0.08, occupancy mean effect size: -0.3). The models did vary on the effects of distance to main roads and percent forest cover. N-mixture showed distance to main roads (mean effect size: -0.09) was moderately significant and the occupancy analysis showed forest cover (mean effect size: -0.32) was moderately significant.

Effects of lynx habitat use also varied between models. The N-mixture model showed distance from urban areas (mean effect size: 0.23, Fig 6) as strongly significant, and distance from dump (mean effect size: 0.13) and distance from main

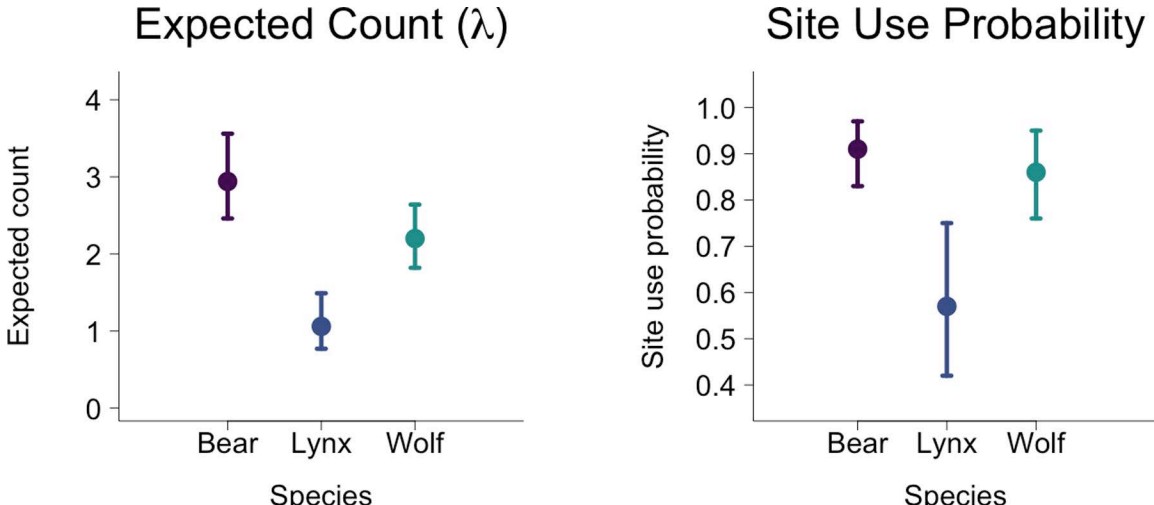

**Fig 2. Differences in expected count (N-mixture) and site use probability (occupancy) for all three species of large carnivores.** The error bar denotes the upper (97.5%) and lower (2.5%) bounds of the confidence interval. Expected count is the derived quantity from the posterior distribution of the N-mixture model, and site use probability is derived from the posterior distribution of the occupancy model.

road (mean effect size: -0.13) as moderately significant. The occupancy model shared distance to urban area (mean effect size: 0.38) as significant, even if only moderately, but also included forest cover (mean effect size: -0.31) and elevation (mean effect size: -0.40) as moderately significant covariates.

## Model comparison

Overall, the N-mixture model was more sensitive than the occupancy model to factors affecting habitat usage across sites. For the three species, N-mixture and occupancy each had ten significant results. However, N-mixture had more strongly significant (Probability of effect > 0.95) results [6] than did the occupancy analysis (2; Tables S1–S3 in the Supplemental files).

Site use intensity (N-mixture) was the highest for bears (mean: 2.94; CI: 2.46–3.56), but not significantly different from wolf site intensity (mean: 2.20; CI: 1.82–2.64). Lynx had the lowest site use intensity (mean 1.06; CI: 0.77–1.49). While bears used the highest percentage of sites (mean: 0.91; CI: 0.83–0.97), it was not significantly different from wolf site use probability (mean: 0.86; CI: 0.76–0.95). However, lynx used a lower percentage of sites (mean: 0.57; CI: 0.42–0.75), as was expected from their assumed lower abundance.

## Discussion

In this study, we used N-mixture and occupancy modeling to understand habitat use across three carnivore species in northeastern Türkiye (Question 1). We also compared the outputs from these models to understand how the results from N-mixture and occupancy analyses differed across species with different abundances (Question 2). This study lays the groundwork for researchers engaged in long-term data collection projects, who need to determine the right model choice for understanding habitat use in their study system. Ultimately, these methods, when used in conjunction, may offer a more complete picture of a species' habitat use. Specifically, we found that assessments of habitat use were sensitive to model type, and possibly to underlying abundance. Therefore, both aspects should be considered when estimating habitat use with remote sensing devices like camera traps.

By combining N-mixture and occupancy analyses, we were able to obtain a better understanding of how environmental and anthropogenic pressures can affect where all three species occur in a landscape, and how they utilize the available

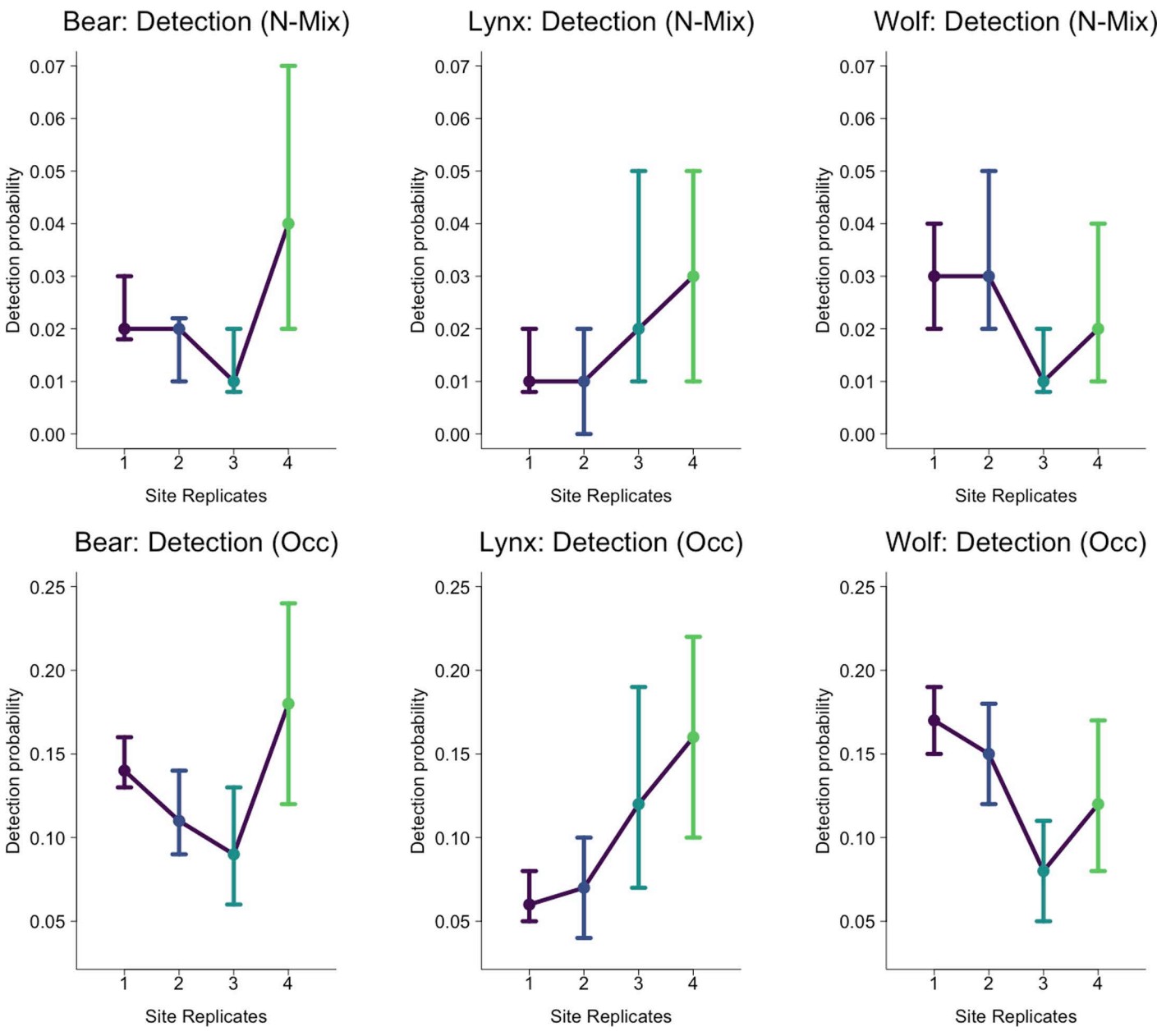

**Fig 3. Difference in detection of all three species across all site replicates for both models. The error bar denotes the upper (97.5%) and lower (2.5%) bounds of the confidence interval. The error bars for several replicates were jittered as they overlapped the mean. Table 2 outlines the exact means and confidence intervals.**

habitat. Within our system, bears have the largest population, but wolves had a similar "expected count" (Fig 2). The pack dynamics and high mobility of wolves may have led to this result even though their actual abundance is lower. Furthermore, this may suggest that wolves use roads more frequently than bears do. This is supported by the differences in their hunting strategies. Lynx had the lowest mean expected count which supports our field observations of these species' underlying abundance throughout the study system. Lynx also had the fewest number of strongly significant habitat

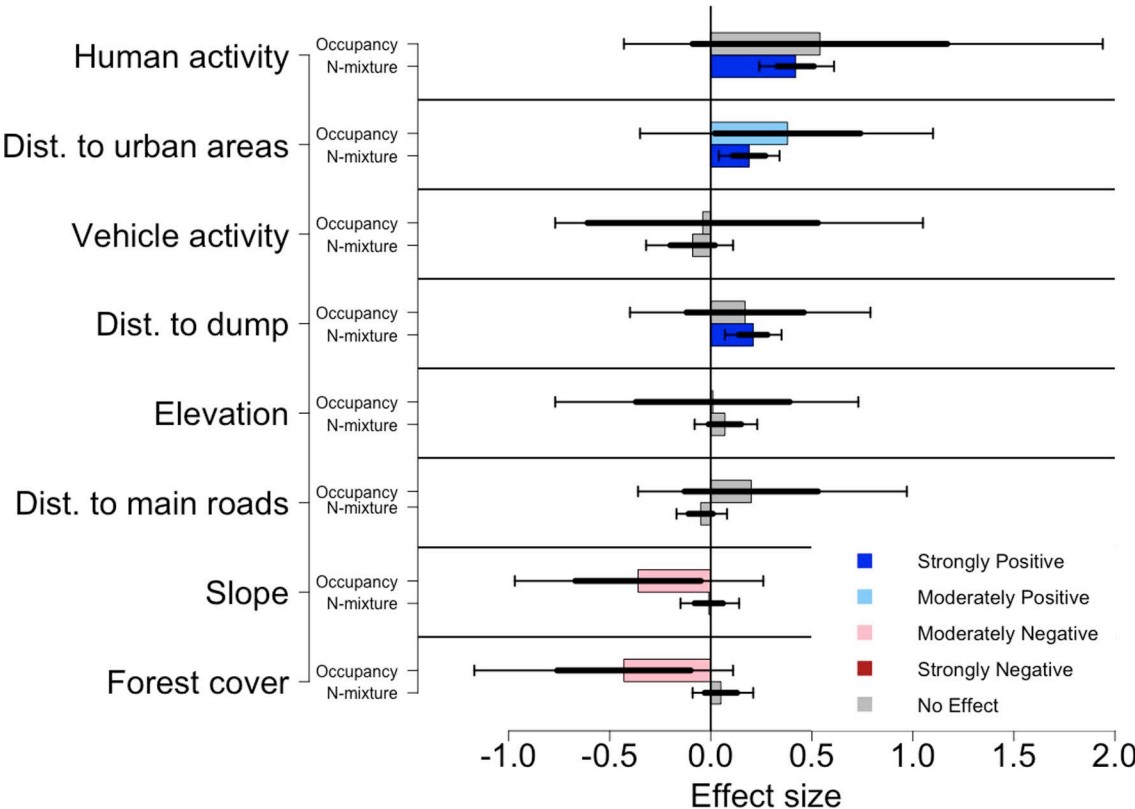

**Fig 4. Effects of environmental covariates on the habitat use of brown bears for N-mixture and occupancy analyses.** Colored bars denote the mean effect size, thicker error bars denote one standard deviation from the sample's mean, while thinner error bars denote the upper (97.5%) and lower (2.5%) bounds of the 95% confidence intervals. Dark blue bars denote a strongly positive response (95% CI), light blue denotes a moderately positive response (85% CI), light red denotes a moderately negative response (85% CI), and dark red denotes a strongly negative response (95% CI).

associations, though we did not explore whether this was due to lower abundance, weaker habitat associations, or lower road use. For all three species, N-Mixture and occupancy models had contradictory results for the effects of the covariates on habitat use. Even though each model had 10 significant results across the three species, only half were shared between models. While none of the shared significant results showed difference in the direction of the effect, several covariates with only one significant effect did (Figs 4–6)

Bears were more likely to occupy locations that were less steep, less forested, and farther from urban centers. However, expected count was most strongly influenced by increases in human activity, followed by increasing distance from the town dump and urban areas (Fig 4). A major dietary resource for bears within the Sarıkamış system is a town dump 6 km outside the town [31,33]. At this dump, they regularly encounter people who come both to utilize the dump and watch the bears [33]. However, the N-mixture model suggested that bears used locations farther from the dump. This went against our initial assumptions of bear habitat use. However, it may be explained by differences between subpopulations of migratory and non-migratory bears in this system. While most of the population is non-migratory and regularly uses the dump [33], roughly 40% of collared bears do not use the dump. Since we do not have cameras in the dump (Blount, unpublished data), and few cameras on roads around the dump, we may have underestimated habitat selection for areas

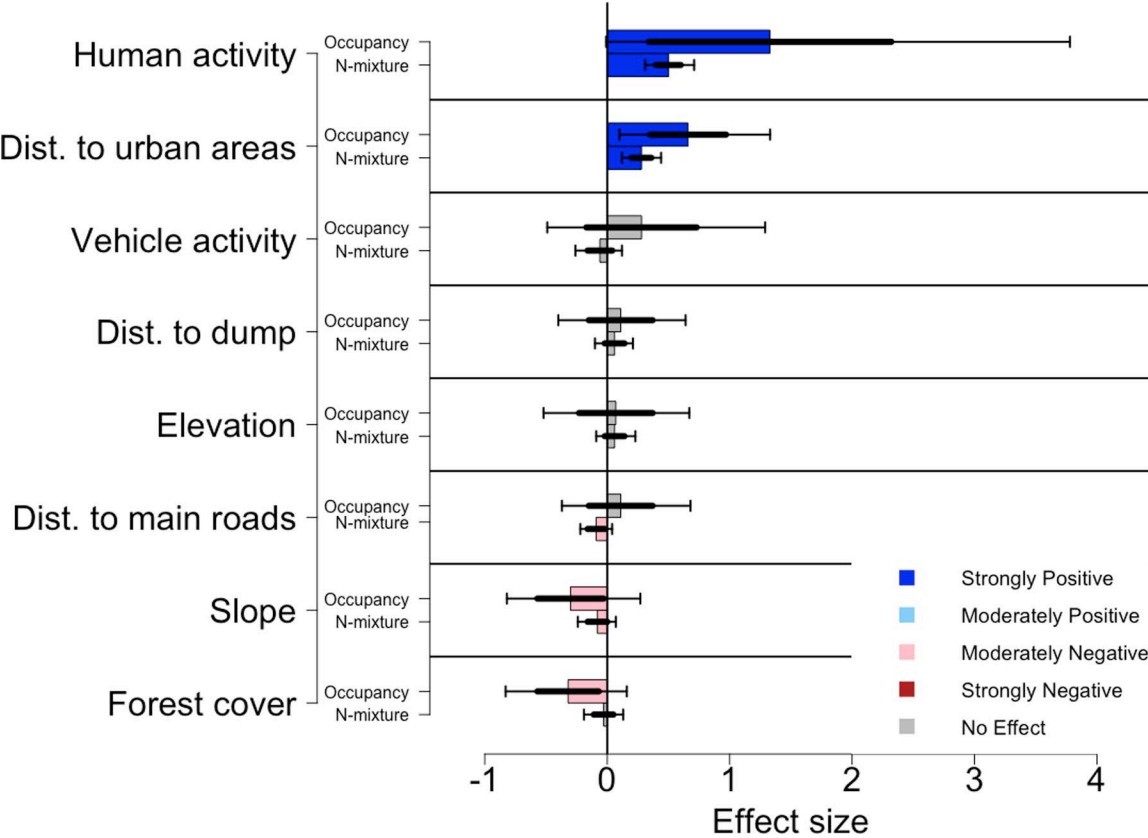

**Fig 5. Effects of environmental covariates on the habitat use of wolves for both N-mixture and occupancy analyses.** Colored bars denote the mean effect size, thicker error bars denote one standard deviation from the sample's mean, while thinner error bars denote the upper (97.5%) and lower (2.5%) bounds of the 95% confidence intervals. Dark blue bars denote a strongly positive response (95% CI), light blue denotes a moderately positive response (85% CI), light red denotes a moderately negative response (85% CI), and dark red denotes a strongly negative response (95% CI).

directly surrounding the dump. Likewise, there may be different preferences for habitat use between subpopulations that get diluted at the population level.

Like bears, wolves had high rates of expected count and site use probability (Fig 3). Wolves were more likely to occupy locations that were flatter, in less forested areas, with higher rates of human activity and that were closer to urban areas (Fig 5). However their site use intensity increased at sites with people, that were closer to urban areas, flatter, and were farther from the main roads. The wolves at our study site have large home ranges (190 km² in the summer and 303 km² in the winter for resident wolves and 784 km² in the summer and 1269 km² in the winter for non-resident wolves. Calculated using 95% KDE; [59]) that makes it necessary for them to use of both forested and unforested sites. In this region, their denning and rendezvous sites, which are their most vital areas, occur mostly within the remaining forest (Blount, Kusak, and Sekercioglu unpublished data). Therefore, wolves may be especially sensitive in their habitat use within the forest interior. Furthermore, the opportunistic hunting style of wolves means they may be making use of open areas, like lightly traveled forest roads, to increase their likelihood of finding prey or scavenging opportunities [60].

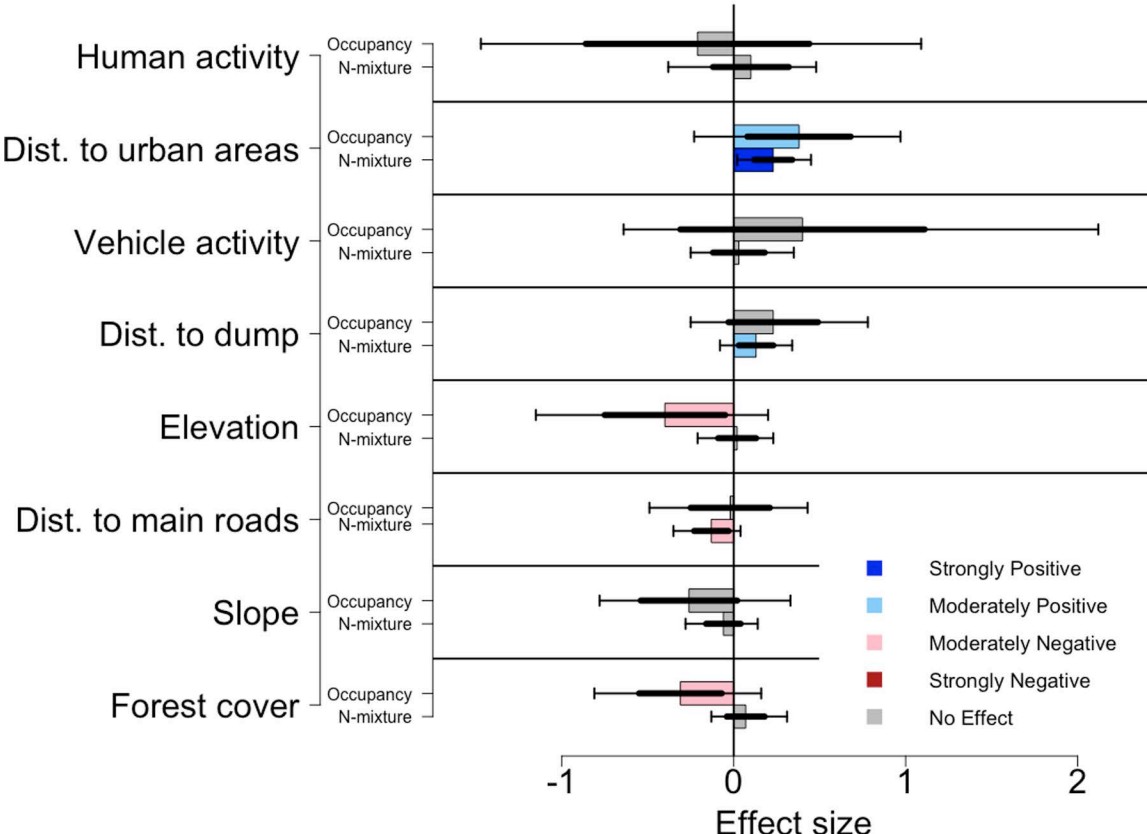

**Fig 6. Effects of environmental covariates on the habitat use of Caucasian lynx for both N-mixture and occupancy analyses.** Colored bars denote the mean effect size, thicker error bars denote one standard deviation from the sample's mean, while thinner error bars denote the upper (97.5%) and lower (2.5%) bounds of the 95% confidence intervals. Dark blue bars denote a strongly positive response (95% CI), light blue denotes a moderately positive response (85% CI), light red denotes a moderately negative response (85% CI), and dark red denotes a strongly negative response (95% CI).

Lynx had the least agreement between the N-mixture and occupancy analyses. This may be due to limited site use probability, detection probability, and expected count or the fact that this species may react differently to what restricts their boundaries and what encourages site use within these boundaries. Occupancy analysis showed that lynx site use was more pronounced in areas closer to human settlements, that were lower in elevation and with limited forest cover. N-mixture results showed lynx favored sites within this range that were close to human settlements and the dump, but farther from main roads. Caucasian lynx are small, shy felids that may compete with bears [61] but typically not with wolves for resources [61, 62]. Therefore, their habitat use may have to be more specific to limit their interactions with both people and their competitors. Activity analysis from this system indicates that all three carnivore species separate temporally from humans to a greater extent than they do from each other [63]. Therefore, less dominant species, such as lynx may have to separate spatially from their competitors to a greater extent. This may cause lynx to be more selective of which habitats they use to reduce risk.

As the two models had limited overlap in covariate significance for habitat use, this study suggests that N-mixture and occupancy modelling may be used in conjunction to provide a better overall understanding of species' habitat usage.

Often, occupancy modelling is used as a detection probability-corrected proxy for abundance or density [15,64,65]. Furthermore, the detection probability parameter in any given occupancy model has also been used as a measure of site abundance or intensity of use [10,20,55]. However, since occupancy modelling utilizes only detection/non-detection data, much of what is gathered from repeat-sampling methods such as camera trapping is condensed and under-utilized. This can limit a study's power of inference, making it difficult to tease out small-scale differences in habitat usage across species and environmental and anthropogenic gradients. For this reason, occupancy modelling is often recommended for elucidating the differences in species' habitat preferences across relatively large scales [66]. In this study, we observed this when comparing the number of "strongly significant" signals produced through occupancy modelling compared to N-mixture modelling, with N-mixture models consistently having more "strongly significant" results. Whereas occupancy modelling can help investigate patterns of habitat usage in creating boundaries or borders, N-mixture modeling may be used to estimate differences in relative activity at the site-specific level. N-mixture modelling utilizes count data, which leverages more information from individual camera sites than binomial detection/non-detection data. This increase in data complexity typically leads to an increase in computational load and a need for larger sample sizes. However, with recent developments in model formulation that control for both over-dispersion and zero-inflation in a Bayesian framework [21], these models may represent a natural alternative to using either the occupancy or detection probability parameter as a proxy for site-level abundance. Therefore, when used in conjunction, occupancy and N-mixture modelling may provide a more thorough picture of how habitats may limit or encourage a species' presence across the landscape. Likewise, using both methods may better illuminate how relatively site-specific correlations of habitat use can affect site use in the immediate microhabitat surrounding individual camera sites.

Importantly, all the data collected for this study came from camera traps that were located on roads, which could potentially bias our habitat use models, as it relates to only how these species utilize locations on roads, and not how they utilize the other parts of their habitat. This highlights the need for further research to understand how road-biased cameras can affect spatial and temporal trends of large carnivores [67,68]. Furthermore, all models used single season, single year assumptions, both of which were violated. We did try to limit the amount of autocorrelation between sites by limiting sites to one deployment a year and accounting for yearly replicates in the models. The consequences of violating these assumptions vary for each model. For N-mixture, violation of these assumptions would be expected to inflate the expected count [69]. For occupancy, the violation of both of these assumptions leads to the violation of the closure assumption, all three of which would inflate the estimation of occupancy [59]. With this in mind, this paper does not try to directly compare the expected count and site use between species. Instead, it looks at the degree at which each habitat covariate influences these parameters across all three species using the same models.

## Conclusions

In this study, we compared habitat use results from N-mixture and occupancy models for three different species of carnivores at different abundances. We found that N-mixture results were more sensitive for all species with respect to highly significant results. We found a high degree (50%) of disagreement between the habitat use conclusions generated by N-mixture and occupancy results. We found that bears, wolves, and lynx all select for locations with higher anthropogenic effects but may separate spatially, which may be due to intraguild competition. Finally, when analyzing habitat use at the local population level, we recommend using a combination of N-mixture and occupancy modeling, as they may be able to provide a complementary understanding of species' habitat use. By combining these techniques, we can leverage the power of each test to better understand what covariates are influencing habitat use. Occupancy analysis can reveal how habitats can create barriers and boundaries for the entire population, which may provide a better understanding of what limits a population's range, while at the finer scale, N-mixture modeling offers more detailed insight into how intensely used certain habitats are within a species' range. Together, they offer a more complete and detection-corrected picture of how a species interacts with its environment.

## Supporting information

**S1 Data. Bear results.**
(XLSX)

**S2 Data. Wolf results.**
(XLSX)

**S3 Data. Lynx results.**
(XLSX)

**S4 Data. Community effects results.**
(XLSX)

## Acknowledgments

ÇHŞ thanks H. Batubay Özkan and Barbara Watkins for their support of the conservation biology research conducted at the Biodiversity and Conservation Ecology Lab at the University of Utah. All authors would like to thank the staff and volunteers of the KuzeyDoğa Society and the Eskişehir Zoo for their support.

## Author contributions

**Conceptualization:** J. David BLOUNT, Austin M. Green, Emrah Çoban.

**Data curation:** J. David BLOUNT, Mark Chynoweth, Emrah Çoban, Josip Kusak.

**Formal analysis:** J. David BLOUNT, Austin M. Green, Mark Chynoweth.

**Funding acquisition:** Çağan H. Şekercioğlu.

**Investigation:** J. David BLOUNT, Emrah Çoban, Josip Kusak, Çağan H. Şekercioğlu.

**Methodology:** J. David BLOUNT, Austin M. Green, Mark Chynoweth, Josip Kusak.

**Project administration:** J. David BLOUNT.

**Resources:** Austin M. Green, Emrah Çoban.

**Supervision:** Josip Kusak, Çağan H. Şekercioğlu.

**Validation:** J. David BLOUNT, Çağan H. Şekercioğlu.

**Visualization:** J. David BLOUNT, Austin M. Green.

**Writing – original draft:** J. David BLOUNT.

**Writing – review & editing:** J. David BLOUNT, Austin M. Green, Mark Chynoweth, Emrah Çoban, Josip Kusak, Çağan H. Şekercioğlu.

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
