## [Decision Letter · Decision Letter 0]

6 Feb 2024

PONE-D-23-35903Combining N-mixture and Occupancy Analysis Offers a More Complete Picture of Carnivore Habitat Selection in Northeastern TürkiyePLOS ONE

Dear Dr. BLOUNT,

Thank you for submitting your manuscript to PLOS ONE. After careful consideration, we feel that it has merit but does not fully meet PLOS ONE’s publication criteria as it currently stands. Therefore, we invite you to submit a revised version of the manuscript that addresses the points raised during the review process.

**ACADEMIC EDITOR:**

The current form of your manuscript needs major revision. There are some issues which must be improved. Please make corrections according to the reviewers' recommendations.

Please submit your revised manuscript by Mar 22 2024 11:59PM. If you will need more time than this to complete your revisions, please reply to this message or contact the journal office at plosone@plos.org . Please include the following items when submitting your revised manuscript:

We look forward to receiving your revised manuscript.

Kind regards,

Karolina Goździewska-Harłajczuk

Academic Editor

PLOS ONE

Journal Requirements:

Additional Editor Comments:

The current form of your manuscript needs major revision. There are some issues which must be improved. Please make corrections according to the reviewers' recommendations.

Reviewers' comments:

Reviewer's Responses to Questions

**Comments to the Author**

1. Is the manuscript technically sound, and do the data support the conclusions?

Reviewer #1: Partly

Reviewer #2: Yes

2. Has the statistical analysis been performed appropriately and rigorously? 

Reviewer #1: No

Reviewer #2: Yes

3. Have the authors made all data underlying the findings in their manuscript fully available?

Reviewer #1: Yes

Reviewer #2: No

4. Is the manuscript presented in an intelligible fashion and written in standard English?

Reviewer #1: Yes

Reviewer #2: Yes

5. Review Comments to the Author

Reviewer #1: This is an interesting study that, based on camera trap data from three top predators, compares the main outcomes of two important analytical methods that have been used to understand patterns of habitat use of animal species (N-mixture and occupancy models). This study also evaluates the effect of distinct detection windows on their parameter estimates. Based on their findings, the authors encourage the joint implementation of both methods to gain a more integral understanding of habitat use. Unfortunately, I see some relatively major problems that must be solved in order to yield a more solid study and numerous relatively smaller issues that also must be fixed.

Major comments

1) Line 70. This hypothesis is not informative. Of course habitat selection must vary among wolves, bears, and lynxes (your current hypothesis is quite obvious), but you need to elaborate on how these inter-specific differences must be. What differences do you expect among these three species in habitat selection? Which landscape features likely facilitate (or prevent) both the presence and large abundance of each species? You must propose a more informative hypothesis based on the biology and ecology of each species.

2) Lines 129-131. A major frailty of this study is that vegetation cover (specifically, tree canopy cover) was not considered as a potentially important predictor of both occupancy and abundance of the three focal species. Several studies in different animal taxa have demonstrated that vegetation cover heavily influences whether a species is present in a particular area (and also likely its site-specific abundance). Thus, why you did not consider this variable as a potential predictor in your occupancy and abundance analyses? I certainly think that bears, wolves, and lynxes prefer sites with abundant vegetation cover. Hence, without this covariate your analyses and results are incomplete and likely misleading. Fortunately, you can obtain quantitative data on vegetation cover using geographic information systems (e.g., satellite images). You could do this for different patch areas around the location of each camera trap (e.g., circles of 0.5, 1, or 2 km in diameter). Without this covariate in your models, I cannot assess the reliability of your main findings or the validity of your conclusions.

3) Lines 144-145 and 150-154. Here, you say that you used a Poisson distribution to build your N-mixture models. Did you verify that this is the appropriate distribution for your three study species? You may also use negative binomial and/or zero-inflated Poisson distributions to build N-mixture models (see Kéry and Royle 2016, Applied Hierarchical Modeling in Ecology, Vol. 1, Academic Press). You must test the appropriateness of these distinct distributions for each focal species. In particular, you say in lines 114-116 that the lynx has the lowest densities based on limited captures on camera traps. Thus, a zero-inflated Poisson distribution could be more appropriate to model the abundance of this species.

4) More details are needed about the different competing models (both N-mixture and occupancy models) that you fitted to the data of all three study species. You should have compared the effects of your distinct predictor variables (and perhaps additive or interactive combinations of predictor variables) on occupancy and abundance. However, in your methods you do not explain if and how you did such comparisons among competing models with distinct environmental predictors. You mention several times in the manuscript that you got a “final model” (for both abundance and occupancy, see lines 157-158, 181-182, and 192) but you fail to explain how did you arrive to such final models. Furthermore, in your results you say that “all N-mixture models had acceptable fit to the data” (line 242) and “all occupancy models also had acceptable fit” (line 284), but it is not clear, at all, which were all these N-mixture and occupancy models. I insist, you must describe clearly all your competing models and how did you select the best predictors for both abundance and occupancy.

5) It is not clear where do your estimates of abundance (figures 2, 3 and 4) and occupancy come from? Did you extract these estimates from your final models for each species? Are they predicted abundances and occupancy probabilities for a site with average values of your environmental covariates? Are they model-averaged estimates (across all competing models)? Please make sure to clarify how exactly did you calculate the main abundance and occupancy estimates that you show in figures 2, 3 and 4.

Additional comments

Abstract

Line 32-33. It is not entirely true that “varying the detection window did not significantly affect modeling outcomes”. According to your results, different detection windows can yield distinct estimates of site-specific abundance of two of your studied species (bears and wolves, see figures 2 and 3). This is a critical result that must be mentioned in the abstract.

Introduction

Lines 41-42. You should mention as well potential effects of features of the landscape such as vegetation cover, availability of water bodies, topography, etc.

Line 50. Insert a space between “habitat selection” and “(13–15)”.

Lines 58-60. The first part of this sentence is awkward and hard to understand. What exactly do you mean by “less sensitive when species are rare and wide-ranging or common”? Are you trying to compare the sensitivity of occupancy models between rare, wide-ranging, and common species? Are occupancy models less sensitive when applied to rare species compared to when applied to common species? Is this what you want to say here? You are mentioning here rare and wide-ranging species but what about species that are rare and geographically restricted? All this is unclear, please reword to clarify.

Lines 63-64. Here you mention that there are a few studies that have compared the results of N-mixture and occupancy models, but you do not provide the references to such studies.

Methods

Line 108. “Lupus” must not be written with capital “L” (i.e., must read “lupus”).

Line 109. A period must be inserted before “While”.

Paragraph that starts in line 119. You must mention in this paragraph what was the minimum distance between cameras and what was the total number of cameras.

Line 124. Insert a space between “3” and “m”.

Line 138. Add pertinent references to this sentence that opens this section.

You write “occupancy” throughout the manuscript using a capital “O”. I think this is incorrect and everywhere it must be written as “occupancy”.

Line 145. “Binomial” must not be capitalized either.

Line 185. You did not define what “Zi,k” denotes in this formula. Please explicitly define this term.

Line 199. Why are you making reference to Table 1 here? This table includes your list of environmental covariates and, in contrast, this sentence is talking about species-specific parameters. This is confusing, please clarify.

Lines 216-219 and elsewhere. You use both uppercase “P” and lowercase “p” to denote P-values. Please be consistent.

Line 220. I would use the Greek symbol chi followed by a superscript 2, instead of spelling out “chi-squared”.

Results

Lines 254-257. This is not entirely true. According to the credibility intervals (CIs) shown in the first panel of figure 3, the 10-day window differs significantly from most other time windows (this CI does not overlap with most other CIs). In addition, the CI range that you report in the text for the 10-day window does not match the CI that can be seen in the first panel of figure 3. In the text, the lower limit is 2.608, whereas in the figure the lower limit is a little higher than 3.5. Please make sure to correct this first panel because it is incomplete and the mean estimate for the 10-day window cannot be seen.

Please make sure that in all your results you write “differences between time windows”, because in some sentences (e.g., lines 254-255 and 257-258) you write “there was no difference” but you do not specify differences between what? Thus, you must always clarify that you are talking about differences between your distinct time windows in parameter estimates.

Line 257. Replace “between” with “in”.

Lines 301-312. In this paragraph, where you present community effects, you make reference to supplementary Tables S1-S3, but the numerical results from your community effects are in Table S4. Please explain or correct this discrepancy.

Line 309. I think “besides” must be replaced with “except for”.

Line 311. Do you mean “survey effort”? In your methods, you described this variable as “effort” (or “survey effort”). Thus, I would use this terminology all throughout the manuscript to avoid confusion. In addition, this variable only affected detection probability, right? (as I mentioned before, you did not describe clearly your candidate models, how you obtained a “final model”, and which covariates were included in this “final model”). If this is the case, you should mention here explicitly that this significant positive effect of survey effort occurred only on detection probabilities.

Line 322. Reword as: “The average mean effect of survey effort on detection probabilities was” (assuming that survey effort only affected detection probabilities, right?).

Lines 345-348. Was human activity really associated positively with occupancy of bears? This is what you say in this text, but in both Table S1 and figure 5 the effect of human activity on occupancy is NEGATIVE in all detection windows. In fact, the average mean effect of human activity that you report in this sentence is -0.18 (line 346). Am I missing something here? Please correct or clarify.

Line 372. Why are you making reference here to figure 3? Shouldn’t this be figure 6 instead?

Lines 394-396. How come the average mean effect of human activity on lynx occupancy was positive (0.14), if all the individual effects (for each time window) of this predictor variable were negative (see figure 7 and Table S3)? Is this a mistake? Please correct or clarify.

Lines 414-415. Here, you must emphasize again that, contrary to site occupancy, the expected counts of bears and wolves did vary among detection windows. This is a critical result.

Discussion

Line 420. Please provide again here the main citations to the N-mixture and occupancy models.

Line 429. Reword as “abundance. Therefore, these three aspects…”

Line 442. Insert a period between “(47-49)” and “Furthermore”.

Lines 449-451. Please remind to the reader here that occupancy models detected fewer environmental effects than N-mixture models.

Line 478. Please make sure to emphasize in this sentence that the effect of vehicles on bears was POSITIVE.

Lines 486-488. These two sentences sound redundant. In both you say that wolves prefer sites farther from the dump and with more vehicle activity. What is the difference between these two sentences? My guess is that in the first one, you are talking exclusively about site occupancy (presence only), whereas in the second you want to say that also in these types of sites (farther from the dump and with more vehicle activity) their abundance is higher. Please reword this text to make this distinction clearer.

Line 491. What is “95% aKDE”?

Line 493. “Rendezvous” is an uncommon term. Please try to use a more common term.

Lines 493-495 and 503-504. Here you are actually highlighting the importance of vegetation cover for wolves and lynxes. Hence, I insist that you must include this variable in your analyses. Otherwise, your ecological story is largely incomplete.

Line 502. “More forest interior” is confusing. Do you mean that lynxes prefer sites FAR from the forest interior or DEEP INSIDE the forest interior? Please reword to make this clear.

Line 511. Replace “dominate” with “dominant”.

Lines 514-515. One important effect of distinct detection windows occurred on the estimated abundances of bears and wolves. I think that is an important result that must be mentioned in this paragraph. Your first sentence in this paragraph minimizes the importance of this effect of detection windows on estimated counts.

Line 529. Replace “counts” with “parameters”.

Conclusions

Lines 535-537. This conclusion is partially wrong because distinct detection windows did have important effects on expected counts of bears and wolves. Please modify it accordingly.

Tables and figures

Line 134. “Occupancy” must not be capitalized.

Table 1. Define what “ESRI” and “CLC” stand for, either in the table heading or as a foot note.

Table 1. Are you sure that “Digital Terrain”, “Model”, and “Camera” must be capitalized? I don’t think so. Please verify.

In all your figures you must explain (in the figure captions) what do the error bars denote. Do they represent credibility intervals (CIs)? If so, are they 95% or 85% CIs?

Captions of figures 2, 3, and 4. Reword as follows: “differences among time windows in N-mixture expected counts, site use probability (occupancy), and detection for…”

Figures 2, 3 and 4. Do not spell out “lambda”. Use instead the Greek symbol lambda.

Figure 3. The first panel is incomplete. The mean estimate of expected counts for the 10-day window cannot be seen.

Captions of figures 5, 6, and 7. Please explain in these captions what do the error bars denote (both thicker and thinner error bars).

Reviewer #2: See attached document. Not sure why there is a minimum character count, given that many people will be submitting their comments as an attached document. I think that this is something that could be raised with PLOS ONE. Limit reached, cheers.

6. PLOS authors have the option to publish the peer review history of their article (what does this mean? ). If published, this will include your full peer review and any attached files.

**Do you want your identity to be public for this peer review?** For information about this choice, including consent withdrawal, please see our Privacy Policy .

Reviewer #1: No

Reviewer #2: **Yes: ** Paolo Strampelli

---

## [Author Response · Author response to Decision Letter 1]

27 Jun 2024

I would like to thank the reviewers for their insightful and precise comments. We have integrated all of the comments, and believe the paper is better because of it. A full response to your concerns has been added.

---

## [Decision Letter · Decision Letter 1]

7 Aug 2024

PONE-D-23-35903R1Combining N-mixture and Occupancy Analysis Offers a More Complete Picture of Carnivore Habitat Use in Northeastern TürkiyePLOS ONE

Dear Dr. BLOUNT,

Thank you for submitting your manuscript to PLOS ONE. After careful consideration, we feel that it has merit but does not fully meet PLOS ONE’s publication criteria as it currently stands. Therefore, we invite you to submit a revised version of the manuscript that addresses the points raised during the review process.

**ACADEMIC EDITOR:**  The most of the comments have been addressed within the current version of the manuscript, however there are still some point which should be revised. Please find the attached file with the additional comments ot the Reviewer.

We look forward to receiving your revised manuscript.

Kind regards,

Karolina Goździewska-Harłajczuk

Academic Editor

PLOS ONE

Additional Editor Comments:

The Authors made many corrections of their manuscript, however there are still some point which should be revised. Please find the attached file with the additional comments ot the Reviewer.

Reviewers' comments:

Reviewer's Responses to Questions

**Comments to the Author**

1. If the authors have adequately addressed your comments raised in a previous round of review and you feel that this manuscript is now acceptable for publication, you may indicate that here to bypass the “Comments to the Author” section, enter your conflict of interest statement in the “Confidential to Editor” section, and submit your "Accept" recommendation.

Reviewer #3: All comments have been addressed

2. Is the manuscript technically sound, and do the data support the conclusions?

Reviewer #3: Partly

3. Has the statistical analysis been performed appropriately and rigorously? 

Reviewer #3: No

4. Have the authors made all data underlying the findings in their manuscript fully available?

Reviewer #3: Yes

5. Is the manuscript presented in an intelligible fashion and written in standard English?

Reviewer #3: Yes

6. Review Comments to the Author

Reviewer #3: See attached review as it is over the character count that I can put in the review box. Here is another sentence as I need a minimum of 100 characters in this box to submit the review.

7. PLOS authors have the option to publish the peer review history of their article (what does this mean? ). If published, this will include your full peer review and any attached files.

**Do you want your identity to be public for this peer review?** For information about this choice, including consent withdrawal, please see our Privacy Policy .

Reviewer #3: **Yes: ** Mason Fidino

---

## [Author Response · Author response to Decision Letter 2]

8 Jan 2025

We would like to thank the reviewer for his review and availability for questions. All responses have been incorperated and detailed in the "response to reviewer" document

---

## [Decision Letter · Decision Letter 2]

25 Feb 2025

Combining N-mixture and Occupancy Analysis Offers a More Complete Picture of Carnivore Habitat Use in Northeastern Türkiye

PONE-D-23-35903R2

Dear Dr. Blount,

We’re pleased to inform you that your manuscript has been judged scientifically suitable for publication and will be formally accepted for publication once it meets all outstanding technical requirements.

Kind regards,

Karolina Goździewska-Harłajczuk

Academic Editor

PLOS ONE

Additional Editor Comments (optional):

All comment have been addressed. The are only some small typos, which should be corrected in the final version of the manuscript.

Reviewers' comments:

Reviewer's Responses to Questions

**Comments to the Author**

1. If the authors have adequately addressed your comments raised in a previous round of review and you feel that this manuscript is now acceptable for publication, you may indicate that here to bypass the “Comments to the Author” section, enter your conflict of interest statement in the “Confidential to Editor” section, and submit your "Accept" recommendation.

Reviewer #3: All comments have been addressed

2. Is the manuscript technically sound, and do the data support the conclusions?

Reviewer #3: Yes

3. Has the statistical analysis been performed appropriately and rigorously? 

Reviewer #3: Yes

4. Have the authors made all data underlying the findings in their manuscript fully available?

Reviewer #3: Yes

5. Is the manuscript presented in an intelligible fashion and written in standard English?

Reviewer #3: Yes

6. Review Comments to the Author

Reviewer #3: I commend the authors on the revisions that they carried out. The revised manuscript is incredibly clear, and the insights gleaned from their analysis. There are a couple of small things that should be corrected, but these are incredibly minor:

Line 218: Both a and x need to be bolded, as they are vectors.

Line 241: Both \psi and \gamma need to be bolded, as they are vectors.

Line 248: There is a comma between \rho and z which should be removed.

Line 308: credible internal instead of confident interval

Line 311 & 318: credible interval instead of confidence interval

Line 352: Typo, I think your fingers went over one spot on the keyboard when trying to type 'on' and instead it has 'pm' in this sentence

I look forward to seeing this paper published, great work on this!

Cheers,

Mason Fidino

7. PLOS authors have the option to publish the peer review history of their article (what does this mean? ). If published, this will include your full peer review and any attached files.

**Do you want your identity to be public for this peer review?** For information about this choice, including consent withdrawal, please see our Privacy Policy .

Reviewer #3: **Yes: ** Mason Fidino

---

## [Editor Report · Acceptance letter]

PONE-D-23-35903R2

PLOS ONE

Dear Dr. BLOUNT,

I'm pleased to inform you that your manuscript has been deemed suitable for publication in PLOS ONE. Congratulations! Your manuscript is now being handed over to our production team.

Kind regards,

on behalf of

Dr. Karolina Goździewska-Harłajczuk

Academic Editor

PLOS ONE